# Genome-Wide Identification and Expression Analysis of the *SHI*-Related Sequence Family in Cassava

**DOI:** 10.3390/genes14040870

**Published:** 2023-04-05

**Authors:** Huling Huang, Jiming Song, Yating Feng, Linling Zheng, Yinhua Chen, Kai Luo

**Affiliations:** 1Sanya Nanfan Research Institute, School of Tropical Crops, Hainan University, Haikou 572025, China; 2Institute of Tropical and Subtropical Economic Crops, Yunnan Provincial Academy of Agricultural Sciences, Baoshan 678000, China

**Keywords:** *SHI*-related sequence, cassava, genome-wide identification, gene expression, abiotic stress

## Abstract

The *SHORT INTERNODES (SHI)*-related sequences (*SRS*) are plant-specific transcription factors that have been quantitatively characterized during plant growth, regeneration, and stress responses. However, the genome-wide discovery of *SRS* family genes and their involvement in abiotic stress-related activities in cassava have not been documented. A genome-wide search strategy was used to identify eight family members of the *SRS* gene family in cassava (*Manihot esculenta* Crantz). Based on their evolutionary linkages, all *MeSRS* genes featured homologous RING-like zinc finger and IXGH domains. Genetic architecture and conserved motif analysis validated the categorization of *MeSRS* genes into four groups. Eight pairs of segmental duplications were detected, resulting in an increase in the number of *MeSRS* genes. Orthologous studies of *SRS* genes among cassava and three different plant species (*Arabidopsis thaliana*, *Oryza sativa*, and *Populus trichocarpa*) provided important insights into the probable history of the *MeSRS* gene family. The functionality of *MeSRS* genes was elucidated through the prediction of protein–protein interaction networks and *cis*-acting domains. RNA-seq data demonstrated tissue/organ expression selectivity and preference of the *MeSRS* genes. Furthermore, qRT-PCR investigation of *MeSRS* gene expression after exposure to salicylic acid (SA) and methyl jasmonate (MeJA) hormone treatments, as well as salt (NaCl) and osmotic (polyethylene glycol, PEG) stresses, showed their stress-responsive patterns. This genome-wide characterization and identification of the evolutionary relationships and expression profiles of the cassava *MeSRS* family genes will be helpful for further research into this gene family and its function in stress response. It may also assist future agricultural efforts to increase the stress tolerance of cassava.

## 1. Introduction

Cassava (*Manihot esculenta* Crantz) is a vital source of food and income for over a hundred million people in tropical countries [1]. Due to its high starch content, it is also considered a promising biofuel crop for ethanol production [2]. According to the The Food and Agriculture Organization (FAO), global cassava production was expected to reach 306 million tons on approximately 34.23 million hectares by 2021 [3]. As cassava is the world’s sixth-largest staple crop [4,5], maintaining high yields is crucial for ensuring food security worldwide. However, biotic challenges such as insect, bacterial, fungal, and viral hazards, as well as abiotic stresses such as water deficit, high salinity, and cold temperatures, can significantly affect plant growth and productivity, including that of cassava. To cope with these adverse conditions, plants have developed a various molecular and physiological defense mechanisms by activating a range of stress-responsive genes. Increasingly, research is focusing on enhancing cassava yields by discovering and charactering key functional genes associated with yield development and environmental adaptation [6]. Although the availability of whole-genome cassava sequences has facilitated the identification of stress-related family genes on a genome-wide scale [7], only a few such genes have been found for cassava breeding so far. Plants show genome-wide transcriptional changes in response to abiotic and biotic stresses, which is critical for their survival by interacting with *cis*-acting elements [8].

The *SHORT INTERNODES* (*SHI*)-related sequence (*SRS*) gene family, also termed as the *SHI* gene family or the *Stylish* (*STY*) family, is a plant-specific transcription factor family that encodes proteins with single zinc finger motifs [9]. The *SRS* family encodes a protein structure with two highly conserved domains: a RING-like zinc finger motif and an IXGH domain. The RING-like region is positioned at the N-terminus, and its amino acid sequence has the following structure: CX_2_CX_7_CX_4_CX_2_CX_7_C, where X denotes variable amino acids [10]. Such a motif has the ability to bind to RNA, protein, and lipid molecules, indicating that it plays a variety of functions in numerous cellular biochemical and physiological processes [11,12]. The IXGH domain at the C-terminus is vital for homodimerization [13]. Certain *SRS* family members, however, may lack sequences encoding the IXGH domain through evolution [10].

*SRS* genes are widely recognized to be involved in biological processes during plant growth and development, particularly plant organ growth, photomorphogenesis, hormone biosynthesis, and environmental stress responses [13,14,15,16,17]. To date, ten *SRS* genes have been discovered in the *Arabidopsis thaliana* genome [18], each of which exhibits both the conventional RING-like zinc finger domain and the IXGH domain, except for *SRS8*, which lacks IXGH structure. The *A. thaliana LATERAL ROOT PRIMORDIUM1* (*LRP1*) gene was the first found plant *SRS* gene, and it was established that the gene is involved in the early phases of lateral and adventitious root primordium formation [19]. *SHI* and the closely related *STY1* and *STY2* genes have been shown to operate jointly to enhance gynoecium, stamen, and leaf growth [18], while *STY1*, *STY2*, *SHI*, or *SRS7* overexpression suppresses stem elongation and the control of tapetum dehiscence [20]. Recent research has found that *SRS5* can adversely influence lateral root development by suppressing *LBD16* and *LBD29* gene expression, while also influencing photomorphogenesis by directly activating *HY5*, *BBX21*, and *BBX22* expression [21,22].

*LjSTY1/2/3* were targets of LjNFYA1, which was critical in nodule differentiation in *Lotus japonicas* (Regel) K. Larsen [23]. *LRP1* works as a transcription factor (TF) in auxin signaling downstream of the Aux/IAA (auxin/indole-3-acetic acid) genes in maize (*Zea mays*) to modulate root growth [24]. In barley (*Hordeum vulgare*), two *SRS* family genes, *LKS2* and *VRS2*, were found to govern awn elongation, pistil shape, and inflorescence patterning [25,26]. Remarkably, a new study indicates that *SRS* family genes may have a role in environmental stress tolerance. Cold and salt stress activate *MeSRS* gene expression in the forage species alfalfa (*Medicago sativa*), indicating that *SRS* genes may play important roles in tissue-dependent signaling pathways [27]. Several stress treatments, including salinity, low temperature, salicylic acid (SA), and methyl jasmonate (MeJA), substantially activated *Melilotus albus SRS* family genes [28]. Soybean (*Glycine max*) *GmSRS18* has been shown to reduce drought and salt tolerance, as well as stress-related gene expression and physiological indicators such as chlorophyll, proline, and relative electrolyte leakage [17]. Due to their important regulatory function, an increasing number of *SRS* genes have been identified in various plant species in recent years, including rice (*O. Sativa*) [29], *Z. mays* [30], *G. max* [17], *M. sativa* [27], *M. albus* [28], and *Phaseolus vulgaris* [31]. Nevertheless, no genome-wide analysis of *SRS* family genes in cassava has been published.

As a result, we performed genome-wide identification and examination of the *SRS* family genes using assembled cassava genomes in our study. A thorough examination of phylogeny, gene/protein structure, conserved motifs, chromosomal placement, synteny, *cis*-element, and expression patterns was carried out. Furthermore, quantitative real-time RT-PCR (qRT-PCR) was used to evaluate the expression patterns of cassava *MeSRS* genes in response to hormone treatments and abiotic stresses. The findings of this study will help to further investigate the activities of the genes in the *MeSRS* family.

## 2. Materials and Methods

### 2.1. Identification of SRS Family Members in Cassava

The genome, protein sequence, coding sequence (CDS), and GFF annotation file of cassava (*M. esculenta* v6.1) were downloaded from EnsemblPlants (http://plants.ensembl.org/index.html, access date: 12 October 2022). The SRS protein sequences from other species, *A. thaliana*, *O. sativa*, *S. oleracea*, *P. trichocarpa,* and *R. communis*, were obtained from the Phytozome v13 database (https://phytozome-next.jgi.doe.gov/, access date: 15 November 2022) or PlantTFDB database (http://planttfdb.gao-lab.org/, access date: 15 November 2022) (Appendix A). The SRS protein sequences from *A. thaliana* and *O. sativa* were used as queries to search the SRS proteins from cassava using BLAST with TBtools software [32]. The screened sequences were then examined for the presence of DUF702 domains by submitting them to Pfam database (http://pfam.xfam.org/family, access date: 20 November 2022), Conserved Domain Database (CCD) (https://www.ncbi.nlm.nih.gov/cdd/, access date: 25 November 2022) [33] and Simple Modular Architecture Research Tool (SMART) (http://smart.embl-heidelberg.de/, access date: 30 November 2022) [34].

### 2.2. Phylogenetic Analysis

Multiple sequence alignments of the identified cassava SRS proteins (MeSRS) and SRS proteins from *A. thaliana*, *O. sativa*, *S. oleracea*, *P. trichocarpa,* and *R. communis* were performed using ClustalW2 with default parameters [35]. The subsequent phylogenetic analysis relied on the neighbor-joining (NJ) method [36], as implemented in the MEGA v7.1 software [37], and a bootstrap analysis was applied based on 1000 replicates [38]. The phylogenetic tree was further beautified using the EvolView web server (http://120.202.110.254:8280/evolview, access date: 29 January 2023) [39].

### 2.3. Characterization of Cassava SRS Genes and Proteins

The protein length, MW, *p*I, and GRAVY of each identified MeSRS protein were calculated using the ProtParam program listed on the Expasy website (https://expasy.org, access date: 12 November 2022) [40]. The subcellular localizations of MeSRS proteins were predicted using the WoLF PSORT web server (https://wolfpsort.hgc.jp/, access date: 12 November 2022) [41]. The MeSRS protein sequences were aligned using DNAMAN ver. 7. The exon–intron structure of the *MeSRS* genes was visualized on the Gene Structure Display Server (GSDS) website (http://gsds.cbi.pku.edu.cn/, access date: 1 December 2022) [42], and the data were obtained from the cassava GFF annotation files. The conserved motifs of MeSRS proteins were predicted using the multiple expectation maximization for motif elicitation (MEME) ver. 5.4.1 online tools (http://meme-suite.org/index.html, access date: 10 December 2022) as follows: the maximum motif number was six, and the others set as default [43]. The secondary structure of the deduced polypeptide was predicted by SOPMA (http://npsa-pbil.ibcp.fr/cgi-bin/npsa_automat.pl?page=npsa_sopma.html, access date: 15 December 2022).

### 2.4. Chromosome Location, Gene Duplication, and Synteny Analysis

Information about the chromosome location of *MeSRS* genes was obtained from the cassava GFF annotation files, and the physical locations of *MeSRS* genes on chromosomes were visualized using TBtools [32]. Gene duplication pairs were identified from the plant genome duplication database (http://chibba.agtec.uga.edu/duplication/index/locket, access date: 15 February 2023). Synteny analysis and chromosomal location diagrams were generated using the Circos program in the TBtools software [32]. The Ka and Ks substitution rates were estimated using the TBtools software [32].

### 2.5. Protein Interaction Network Diagram and Cis-Regulatory Element Analysis

The functional relationship of the MeSRS proteins was predicted using the STRING protein interaction database (http://string-db.org/, access date: 20 November 2022) [44,45]. To analyze the regulatory elements in the *MeSRS* genes, the TBtools software was used to extract the 2000 bp upstream sequences of the *MeSRS* coding DNA sequences from the cassava genome data [32]. Possible *cis*-acting elements were identified using the PlantCARE database (http://bioinformatics.psb.ugent.be/webtools/plantcare/index.html, access date: 29 November 2022) [46].

### 2.6. In Silico mRNA Levels of MeSRS Genes in Different Tissues

The transcriptome data were retrieved from the NCBI database (accession number: GSE82279) submitted by Wilson et al. [5] and used to investigate the expression profiles of *MeSRS* genes in different tissues, including leaves, mid veins, lateral buds, somatic embryos (OES), friable embryogenic callus (FEC), fibrous roots (FR), storage root (SR), stems, petioles, root apical meristem (RAM), and shoot apical meristem (SAM). Fragments per kilobase of transcript per million reads mapped (FPKM) values were calculated to evaluate the gene expression. Log^2^-transformed heatmaps of all *MeSRS* genes were produced using TBtools software [47].

### 2.7. Plant Materials and Stress Treatments

South China 9 (SC9) cassava, a common edible cultivar in China that has good performance in terms of properties of starch and stress tolerance, was used in this research. The stems, approximately 15 cm in length with two to three buds, were subcultured for approximately 30 days on Murashige and Skoog (MS) medium at 25 °C under a 16 h light/8 h dark cycle at Hainan University (Haikou, Hainan, China). Uniform seedlings were chosen to study the transcriptional response of cassava *MeSRS* genes under abiotic stresses and phytohormone treatments. The seedlings were transferred to MS medium containing 20% PEG, 400 mM NaCl, 100 μM SA, and 100 μM MeJA, respectively, and four treatment time points (0, 4, 12, and 24 h) were selected. Leaf samples were collected with three replications per treatment at each time point. All samples were flash-frozen in liquid nitrogen and stored at −80 °C until use.

### 2.8. RNA Isolation and qRT-PCR

RNAprep Pure Plan Plus Kit (TIANGEN Biotech Co., Ltd., Beijing, China) was used to isolate RNA from the leaves according to the manufacturer’s instructions. Total RNA (1 ug) was used for first-strand cDNA synthesis using a reverse transcriptase kit (M1631, Thermo Fisher Scientific, Waltham, MA, USA). The elongation factor 1 α (*EF1α*) gene was used as an endogenous control. Quantitative primers were designed using the website Primer3 (http://bioinfo.ut.ee/primer3-0.4.0/) based on the identified *MeSRS* gene sequences, and the primers used in this study are listed in Appendix A. Real-time PCR was performed on a 7500 Real Time PCR System (Thermo Fisher Scientific) with a total reaction volume of 20 μL, containing 2 μL cDNA template, 1 μL forward primer, 1 μL reverse primer, 10 μL qPCR Master Mix, and 6 μL sterilized ddH_2_O. The real-time PCR amplification protocol was set at 95 °C for 30 s for 40 cycles, 95 °C for 5 s, 55 °C for 30 s, 72 °C for 30 s, and 72 °C for 10 min [48]. Relative fold expression changes of the eight *MeSRS* genes were normalized to the *EF1α* gene and calculated using the 2^−ΔΔCt^ method described by Schmittgen and Livak [49]. Each sample was analyzed in three independent biological replications. Differences between groups were analyzed using Duncan’s multiple range tests with SPSS Statistics ver. 20.0 (IBM, Armonk, NY, USA), and a *p* ≤ 0.05 was considered statistically significant.

## 3. Results

### 3.1. Identification and Phylogenetic Analysis of MeSRS Genes in Cassava

The cassava genome revealed eight non-redundant *SRS* family members (*MeSRS1* to *MeSRS8*) (Table 1). Figure 1 shows a comprehensive sequence alignment of the MeSRS proteins. All of the MeSRS proteins had the conserved RING-like zinc finger domain CX_2_CX_7_CX_4_CX_2_CX_7_C and the IXGH domain, which are basic features of the *SRS* gene family (Figure 1). Additionally, the physicochemical characteristics of these *MeSRSs* were investigated. The CDS lengths for *MeSRS* genes ranged from 642 bp (*MeSRS7*) to 1080 bp (*MeSRS2*), while the sizes of the MeSRS proteins ranged from 213 a.a (MeSRS7) to 359 a.a (MeSRS2). The molecular weights (MWs) ranged from 23.05 kDa (MeSRS7) to 37.22 kDa (MeSRS2), and the projected isoelectric points (*p*Is) ranged from 6.13 (MeSRS6) to 8.98 (MeSRS7). All of the MeSRS proteins had a negative grand average of hydropathicity (GRAVY), suggesting that they were hydrophilic. The WoLF PSORT web service was used to identify MeSRSs’ subcellular localizations. Except for *MeSRS2*, which was predicted to be localized in the endoplasmic reticulum, almost all *MeSRSs* were predicted to be localized in the nucleus (Table 1). The SPOMA online program was employed to estimate the structural features of eight MeSRS proteins to better comprehend their molecular function (Appendix A). According to our findings, the random coil is the primary component of the secondary structure of MeSRS proteins, accounting for more than 60% of all MeSRS proteins. The secondary structure α helix of MeSRS1 and MeSRS3 proteins, on the other hand, accounted for a substantial amount (more than 17%), whereas the portion of other proteins was minimal, with the lowest at 8.36% for *MeSRS2*. In the MeSRS protein structure, the fraction of extended chains was less than 18%.

An unrooted phylogenetic tree was generated using MAGA 7.1 software by utilizing multiple sequence alignment of SRS protein sequences from six distinct plant species, including eight in cassava, ten in *A*. *thaliana*, six in *O. sativa*, four in *Spinacia oleracea* (*S. oleracea*), ten in *Populus trichocarpa* (*P. Trichocarpa)*, and five in *Ricinus communis* (*R. Communis)* (Appendix A). Based on their evolutionary relationships, we grouped the 43 SRS proteins into six different groups (I-VI) (Figure 2). Group VI had the most SRS proteins, with nine from all six species, whereas Group IV only had three from monocot plants. The eight *MeSRS* genes were divided into five groups: I, II, III, V, and VI, each with one, two, two, one and two members, respectively. *MeSRSs* are more strongly connected to *SRSs* from dicots, particularly *R. communis* and *P. trichocarpa* (Figure 2).

### 3.2. Gene Structure and Conserved Motifs of MeSRSs

Figure 3A depicts a distinct cluster of cassava SRS proteins. The results showed that the tree’s structure matched that of the phylogenetic tree constructed using *SRS* sequences from the six plant species displayed in Figure 2. A detailed comparison of exon–intron structure was performed to further understand the structural properties of *MeSRS* genes (Figure 3B). Six *MeSRS* genes (*MeSRS1*, *MeSRS2*, *MeSRS3*, *MeSRS4*, *MeSRS5*, and *MeSRS7*) had two exons and one intron, but two *MeSRS* genes in group II (*MeSRS6* and *MeSRS8*) had three exons and two introns. Exon–intron organization was generally comparable within the same group, supporting their tight evolutionary ties (Figure 3B). The MEME online program discovered six motifs to further investigate structural diversity and anticipate the function of the MeSRS protein (Figure 3C). The preserved motif lengths ranged from 18 (motif 3) to 50 (motif 2) a.a., while their logo symbol is illustrated in Figure 3D. A RING-like zinc finger motif CX_2_CX_7_CX_4_CX_2_CX_7_C appears in motif 1, while the IXGH conservation domain exists in motif 2. Motifs 1 and 2 were found in all MeSRS proteins, which is in line with the findings of multiple alignment of sequence analyses (Figure 1). It is worth mentioning that several conserved motifs were discovered in particular groups. For example, motif 3 was found only in groups VI and V, motifs 4 and 5 in groups III, IV, and VI, and motif 6 in two MeSRSs from group I. (Figure 3C). This finding implies that the distinct roles of each subgroup may be linked to certain motifs.

### 3.3. Chromosome Localization, Gene Duplication and Synteny Analysis

The eight *MeSRS* genes were spread across six cassava chromosomes. Two genes were found on chromosomes 1 and 2, and one gene on chromosomes 4, 5, 11, and 18. (Table 1; Figure 4A). We conducted a collinear analysis of *MeSRS* genes to better understand their evolutionary history. As segmental duplication genes, a total of eight homologous gene pairs including seven *MeSRS* genes were found. The duplicated pairings are *MeSRS1*: *MeSRS8*, *MeSRS1*: *MeSRS3*, *MeSRS1*: *MeSRS6*, *MeSRS2*: *MeSRS4*, *MeSRS3*: *MeSRS*6, *MeSRS7*: *MeSRS8*, *MeSRS8*: *MeSRS3*, and *MeSRS8*: *MeSRS6* (Figure 4B). These findings suggested that these genes were perhaps created by segmental duplication events, which operated as a primary driving factor in the development of the *MeSRS* family. Furthermore, the non-synonymous (Ka) and synonymous (Ks) substitution rates were calculated for each homologous gene pair using the TBtools program. All of the *MeSRS* orthologous gene pairs showed a Ka/Ks ratio < 1 (Table 2). These findings point to the *MeSRS* genes’ conserved evolution. We created three comparative syntenic maps of cassava compared to three representative species, including two dicot plants, *A. thaliana* and *P. trichocarpa*, and one monocot plant, *O. sativa*, to further infer the evolutionary pathways of the *MeSRS* family (Figure 4C). Therefore, we identified pairwise homologues of the *MeSRS* genes in *A. thaliana*, *P. trichocarpa*, and *O. sativa* to detect 16, 27, and 2 pairs of homologous genes, respectively (Appendix A).

### 3.4. Identification of Cis-Acting Elements in the Promoters of MeSRS Genes

PlantCARE was used to conduct a *cis*-elements study on the 2000 bp area upstream of the *MeSRS* genes (Figure 5). Each component possesses its own unique sequence and serves a specific function. In this study, eighteen *cis*-acting elements were identified: eight stress-sensitive (MYB, MYC, TC-rich repeats, MBS, STRE, W box, WUN-motif, and LTR), ten hormone-responsive (ABRE, CGTCA motif, ERE, GARE-motif, P-box, TATC-box, TCA-element, TGACG-motif, AuxRR-CORE, and TGA-element) *cis*-acting elements, and nine growth and development related (ARE, CAT-box, as−1, O2-site, circadian, HD-Zip 1, MSA-like, OCT, and GC-motif) *cis*-acting elements. ERE and ARE were discovered in each *MeSRS* gene promoter. *MeSRS3* is expected to have the largest amount of *cis*-elements (Figure 5). Altogether, the promoter regions of each of these genes include a minimum of three types of stress-responsive and four types of hormone-responsive *cis*-acting elements, indicating that *MeSRS* genes may be engaged in abiotic and biotic stress responses.

### 3.5. Protein–Protein Interaction Network and GO Annotation Analysis

Utilizing protein–protein interaction network to connect unknown functional proteins can help us comprehend the many biological activities of proteins and the dynamic regulatory networks that exist among distinct biomolecules. In this study, the functional and physical interactions among potential MeSRS proteins were examined using the STRING program based on the *A. thaliana* association model (Figure 6). Each node represents all proteins generated by a single protein-coding locus. The predicted number of boundaries and the average local clustering coefficient were 10 and 0.313, respectively, and the *p*-value for protein–protein interaction enrichment was 8.29e-11. We found nine SRS functional molecules (*SHI*, *STY1*, *STY2*, *LRP1*, *SRS3*, *SRS4*, *SRS5*, *SRS6*, and *SRS7*) and five putative interaction proteins (KIN13A, NGA3, YUC4, AT1G32730, and AT3G06840) directly connected to MeSRS proteins using statistical analysis of eight MeSRS proteins (Figure 6). Among the seven discovered similar proteins in *A. thaliana*, *SRS3* had the most interaction partners, including KIN13A, YUC4, AT1G32730, and AT3G06840.

The gene ontology (GO) functional analyses were carried out utilizing the PANNZER web server to identify the biological process, molecular function, and cellular component characteristics of the MeSRS proteins (Appendix A). In the “biological process” category, auxin biosynthesis process (GO:0009851), auxin-activated signaling pathway (GO:0009734), regulation of DNA-templated transcription (GO:0006355), and positive regulation of nucleic acid-templated transcription (GO:1903508) characteristics of the MeSRS proteins were revealed. In the “molecular function” category, all proteins were classified as DNA-binding transcription factor activity (GO:0003700) and DNA binding (GO:0003677). Two GO terms related to cellular components, including nucleus (GO:0005634) and membrane (GO:0016020), were specifically enriched. These findings show that MeSRS proteins have a variety of roles in cell metabolism.

### 3.6. Expression of MeSRS Genes in Different Tissues

Wilson et al. [5] made their cassava RNA-seq data publicly available, and their dataset was used to investigate the expression patterns of *MeSRS* genes across eleven tissues. All eight *MeSRS* genes were included in the hierarchical clustering analysis (Figure 7) and found to be expressed differently across different cassava tissues. *MeSRS2* and *MeSRS5* exhibited higher expression levels in almost all tissues than the other *MeSRS* genes, suggesting that they may play important roles in cassava development. In contrast, *MeSRS7* expression was low in all tissues tested. Some *MeSRS* genes exhibited tissue-specific expression patterns. *MeSRS1*, *MeSRS6*, and *MeSRS8* exhibited higher expression levels in friable embryogenic callus (FEC), organized embryogenic structures (OES), fibrous roots (FR), and lateral buds than in other tissues. *MeSRS4* had the greatest expression in storage root (SR). The high levels of expression of these *MeSRS* genes in a specific tissue suggest that they may play crucial roles in tissue growth or functionality (Figure 7).

### 3.7. Response of MeSRS Genes to Hormone Treatments and Abiotic Stresses

To study the possible functions of the cassava *MeSRS* genes in response to various stimuli, qRT-PCR was used to examine the expression patterns of these genes in cassava seedlings treated with salicylic (SA), methyl jasmonate (MeJA), NaCl, and polyethylene glycol (PEG) (Figure 8 and Figure 9). As a control, the expression level at 0 h was set as the baseline (CK). Except for *MeSRS3*, the seven *MeSRS* genes were strongly activated or repressed by SA treatment. *MeSRS1* and *MeSRS2* were suppressed after 12 h and 4 h, respectively. *MeSRS4* and *MeSRS6* demonstrated a similar dynamic, with upregulation beginning at 12 h. *MeSRS5* and *MeSRS7* were considerably upregulated at 12 h, but *MeSRS8* showed no significant change from 0 to 12 h before being significantly upregulated at 24 h. *MeSRS1* and *MeSRS7* levels increased progressively under MeJA therapy and peaked at 24 h. *MeSRS2* was significantly activated and sustained reasonably high levels of expression. *MeSRS3* and *MeSRS5* had a similar expression profile, with a strong increase after 12 h and a subsequent decrease after 24 h. *MeSRS4 and MeSRS6* showed a similar expression profile, whereas *MeSRS6* remained relatively low at 24 h. *MeSRS8* exhibited almost no change during the first phase of MeJA treatment and began to be upregulated after 24 h (Figure 8).

Following NaCl treatment, the expression of *MeSRS1*, *MeSRS2*, *MeSRS5*, *MeSRS7*, and *MeSRS8* increased considerably at 4 h or 12 h and then decreased after the peak time point. *MeSRS3* was progressively suppressed, and its expression was downregulated at 24 h compared to 0 h. *MeSRS4* levels were much lower after 12 h, then rose after 24 h and eventually peaked at 24 h. *MeSRS6* was shown to be considerably up-regulated after 24 h (Figure 9). The expression of most *MeSRS* genes was considerably affected by PEG treatment, except for *MeSRS7*, whose expression remained relatively steady at all time periods. *MeSRS3*, *MeSRS5*, and *MeSRS6* levels first increased but subsequently declined, peaking at 4 h or 12 h. *MeSRS1*, *MeSRS4*, and *MeSRS8* all showed similar “down-up-down” tendencies from 0 h to 24 h. *MeSRS3* was significantly upregulated at 24 h compared to 0 h, with a continuous rise in expression (Figure 9).

Taken together, most *MeSRS* genes were significantly affected by a variety of treatments. It was hypothesized that different *MeSRS* genes had diverse roles and used different coping techniques in response to stress or phytohormone signals in cassava.

## 4. Discussion

*SRS* is a plant TF family that has an important regulatory function in plant development and growth. In *A. thaliana*, *SRS* family genes have been well demonstrated to play vital roles in regulating photomorphogenesis [21,22], floral organ development [14,18], lateral root formation [17,21], carpel development [15], and phytohormone response [10,16]. Recent studies have further confirmed the involvement of *SRS* genes in abiotic stresses and phytohormone response [17,27,28,50], and their potential utilization value for genetic improvement of crop resistance. Although studies have been performed on *SRS* genes in several plant species [17,27,28,29,30,31], no systematic research on the *SRS* gene family in cassava has been reported. It is necessary to investigate the possible role of *SRS* in cassava. In this study, eight *SRS* genes were discovered in the cassava genome and named *MeSRS1* through *MeSRS8* based on their chromosomal placement (Table 1; Figure 4). A higher number of *SRS* genes were found in the cassava genome than in *O. sativa* (*n* = 6) [29], but fewer than in *A. thaliana* (*n* = 10) [18,51], *P. trichocarpa* (*n* = 10) [52], *Z. mays* (*n* = 11) [30], and *G. max* (*n* = 21) [17]. The number of *SRS* family genes between species may be related to gene duplication events and genome size [53].

The phylogenetic analysis of 43 SRS proteins from *A. thaliana*, *O. sativa*, *S. oleracea*, *P. trichocarpa*, and *R. communis* allowed us to analyze the evolutionary history and estimate the recurring occurrences during the growth of the *SRS* family. SRS proteins from six species were classified into six classes, with group VI having the most members (10) and being present in all six investigated species. Group IV was a monocot-specific group, whereas groups I, II, III and V had dicot members (Figure 2), which was consistent with He et al. [30]. It has been demonstrated that the four SRS protein subfamilies evolved through duplication of a common ancestor prior to the divergence of the monocot and eudicot lineages. The results of phylogenetic trees suggest that the grouping link is more likely when the functions are comparable [54]. Eight *MeSRS* genes, for example, are derived from groups I, II, III, V and VI, which have closer phylogenetic relationships and genetic structures with *A. thaliana* (Figure 2 and Figure 3A). This implies that *MeSRS* members could be operationally redundant and regulate auxin homeostasis in a dose-dependent manner during plant development [18]. *SRS* genes in plants had highly conserved exon/intron architectures. Exons in *MeSRS* genes in cassava vary from 2 to 3 (Figure 3B), which is comparable to *O. sativa* (2) [29], *Z. mays* (2–3) [30], *G. max* (2–4) [17], and *M. sativa* (2–4) [27]. We found six conserved motifs that helped us understand structural similarities among *MeSRS* proteins (Figure 3C; Figure 3D). Motif 1 was linked to RING-like region domains, whereas motif 2 was linked to IXGH domains. Other motifs were either increased or removed and shifted, which might be connected to the distinct genetic variety created by natural selection in the face of varied adversities in cassava (Figure 3D). This also suggested that the MeSRS proteins might have different functions.

The major driving mechanisms for gene family expansion in diverse plant species are gene duplication events, including segmental, tandem, and whole-genome duplications. Genetic mapping of *MeSRS* genes found that eight *MeSRS* genes were distributed across six chromosomes, with the number of genes ranging from one to two on each chromosome (Figure 4A). Analysis of gene duplication events identified eight pairs of duplicated genes with strong nucleotide sequence similarity. Furthermore, both Ka/Ks ratios were lower than one, suggesting that all duplicated *MeSRS* gene family members were under purifying selection pressure after gene duplications (Figure 4B; Table 2). In *M. albus SRS* genes, eight pairs of segmental duplication genes were identified, with gene duplication playing a key role in gene family expansion [28]. Two segmental duplications of *OsSRS* genes were observed in *O. sativa* [29]. Additionally, synteny analysis showed at least 16 orthologous gene pairs between cassava and the other two dicot species, *P. trichocarpa* and *A. thaliana*, but only two orthologous gene pairs between cassava and the monocot species *O. sativa*, suggesting that these gene pairs emerged after the divergence of dicot and monocot plants (Figure 4C). The phylogenetic tree also supports this hypothesis, demonstrating that cassava was more closely linked to *P. trichocarpa* than to *A. thaliana* and *O. sativa* (Figure 2).

A *cis*-acting regulatory element is a non-coding DNA segment found in the promoter region of genes. Drought sensitive (MBS, MYB), low temperature responsive (LTR), and abscisic acid (ABA) responsive (ABRE) *cis*-regulatory elements were found in various *MeSRS* promoters (Figure 5), showing that these genes might be activated by ABA and other abiotic stimuli [55,56]. Similarly, MYC responds to salt, drought, and ABA stresses by forming complexes with MYC-binding proteins [55,57,58]. In addition, three *MeSRS* genes had a W-box element (Figure 5), which is the DNA binding site of the WRKY protein produced by salicylic acid and has a role in disease defense [59,60,61]. Phytohormones are also required for plant response to abiotic stresses [62]. In this work, we determined that the *MeSRS* gene contains a variety of hormone response elements, including ethylene (ERE), gibberellin (TATC-box), MeJA (CGTCA/TGACG-motif), and IAA (TGA-element) (Figure 5). The *cis*-acting elements analysis suggests that the function of the *MeSRS* genes may play a role in abiotic and biotic stress responses, which is consistent with the findings of *SRS* gene family research in *G. max* [17].

Phytohormones and environmental stresses have a significant impact on crop development and growth. Previous research found that TFs play a role in many plants’ abiotic and biotic stress responses [63,64,65,66]. Plants contain putative stress or phytohormone-related *SRS* genes [28,30]. A recent study also discovered that *GmSRS18* is present in the cell nucleus of *G. max* and can improve the susceptibility of transgenic *A. thaliana* to drought and salt stresses [17]. In this investigation, qRT-PCR was utilized to confirm the expression level of *MeSRS* genes under different conditions of phytohormone induction (SA and MeJA), osmotic pressure (PEG), and salt stress (NaCl) (Figure 8 and Figure 9). *MeSRS* genes responded differently to various hormone treatments. For example, *MeSRS1* and *MeSRS2* may be inhibited by SA but stimulated by MeJA (Figure 8), implying that SA and MeJA trigger antagonistic pathways. Almost all *MeSRS* genes were upregulated following PEG and NaCl treatments (Figure 9), suggesting that they play a role in the regulation of diverse stress-related responses. Interestingly, the changes in gene expression and time across various treatments are not simple linear connections, indicating that the regulation of distinct *MeSRS* family members is a complex process, as corroborated by other studies [28,47]. Although our findings suggest that the *MeSRS* genes respond to phytohormone stimulation and abiotic stress, much more research is required to fully understand this response mechanism.

In all, we identified eight non-redundant *MeSRS* genes in the cassava genome. A comprehensive genome-wide investigation of the *MeSRS* gene family was carried out, including phylogenetic analysis and gene architecture, chromosomal position, evolutionary traits, and a functional network. Following that, the *cis*-elements in the promoter region were studied, as well as the expression patterns in different tissues and in response to diverse stimuli, using transcriptome or qRT-PCR analysis. Our findings will be valuable for future functional characterization of *MeSRS* genes in cassava, and for improving the adaptive ability of plants exposed to environmental challenges.

## Figures and Tables

**Figure 1 genes-14-00870-f001:**
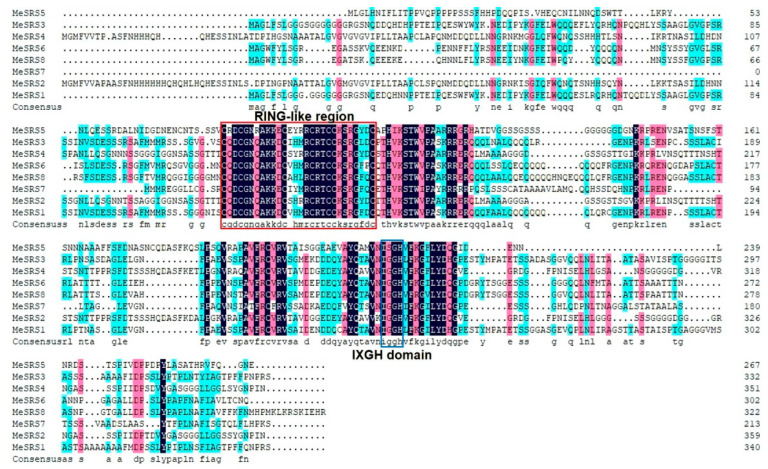
Multiple sequence alignments of MeSRS proteins. The RING-like region (CX_2_CX_7_CX_4_CX_2_CX_7_C) and the IXGH domain are marked by the red and blue boxes, respectively.

**Figure 2 genes-14-00870-f002:**
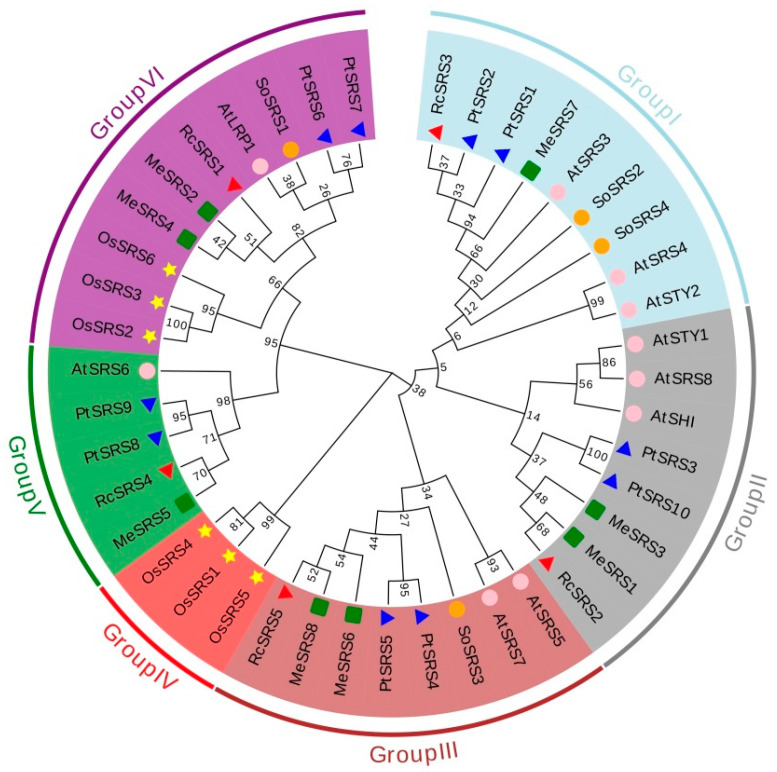
Phylogenetic analysis of SRS proteins in six plant species. Unrooted phylogenetic tree of plant SRS proteins was constructed using the neighbor-joining (NJ) method with 1000 bootstrap value. Different groups were distinguished by different colors, and the SRS proteins from different plant species were marked by different color and shape. Me, *Manihot esculanta* Crantz; At, *Arabidopsis thaliana*; Os, *Oryza sativa*; So, *Spinacia oleracea*, Pt, *Populus trichocarpa*; Rc, *Ricinus communis*. The accession numbers and protein sequences of all *SRS* genes used in the present study are listed in Appendix A.

**Figure 3 genes-14-00870-f003:**
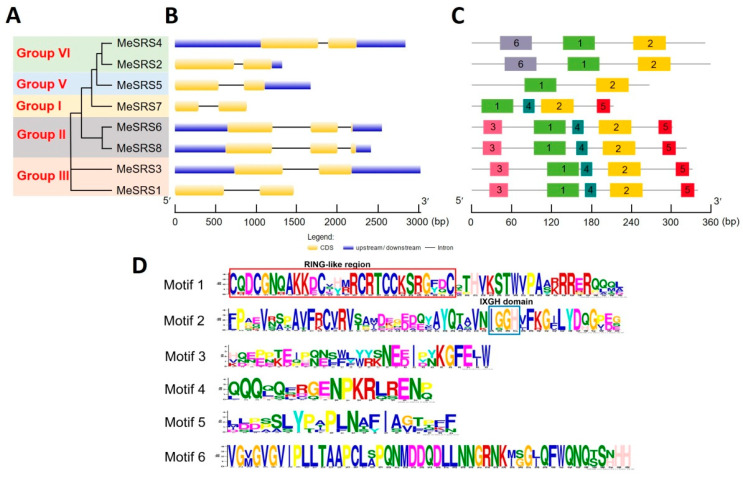
Phylogenetic relationship, gene structure, and protein motif analysis of MeSRS family members. (**A**) Phylogenetic tree of MeSRS proteins was constructed by the neighbor-joining (NJ) method with 1000 bootstrap values. (**B**) Exon–intron organization of *MeSRS* genes, using the GSDS program. CDS, upstream/downstream and introns are shown. (**C**) Distribution of the conserved motifs. Conserved motifs in the *MeSRS* proteins were identified by MEME software. Gray lines represent the non-conserved sequences, and different colored boxes represent each motif. (**D**) The logo map of conserved sequences of six putative motifs of the MeSRS protein. The conserved RING-like region (CX_2_CX_7_CX_4_CX_2_CX_7_C) and the IXGH domain are marked by the red and blue boxes, respectively.

**Figure 4 genes-14-00870-f004:**
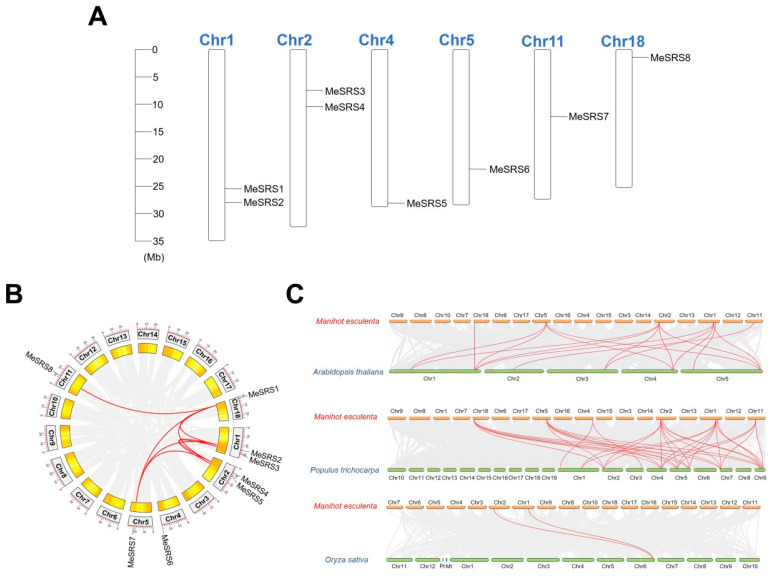
Mapping and synteny analysis of *MeSRS* genes. (**A**) Illustrative diagram of chromosomal distribution of *MeSRS* genes. The chromosome number is indicated above each bar, and the scale on the left is in megabases (Mb). The chromosome size is indicated by its relative length using the information from GFF3. (**B**) CIRCOS figure of *MeSRS* genes. The gray line in the background indicates a collinear block in the genome of cassava, while the red line highlights the isomorphic gene pair. The chromosome number is indicated in each chromosome. (**C**) Syntenic relationships between *MeSRS* genes in cassava with other *SRS* genes in three other representative plant species. Gray lines in the background indicate the collinear blocks within cassava and other plant genomes. Red lines in highlight indicate the syntenic *SRS* gene pairs.

**Figure 5 genes-14-00870-f005:**
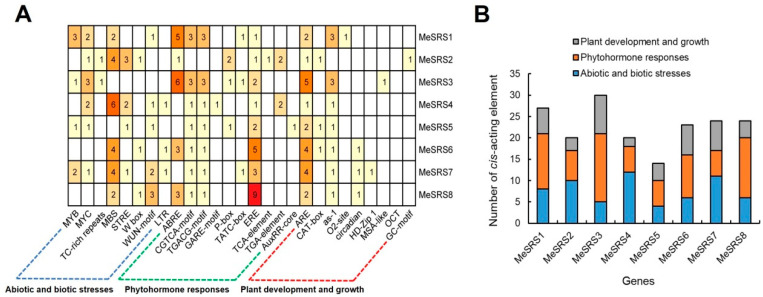
Analysis of *cis*-acting elements in *MeSRS* genes. (**A**) Different colors and numbers on the grid indicate numbers of different *cis*-acting elements in each *MeSRS* gene. (**B**) Different colors on the histograms represent the sum of the *cis*-acting elements in each category.

**Figure 6 genes-14-00870-f006:**
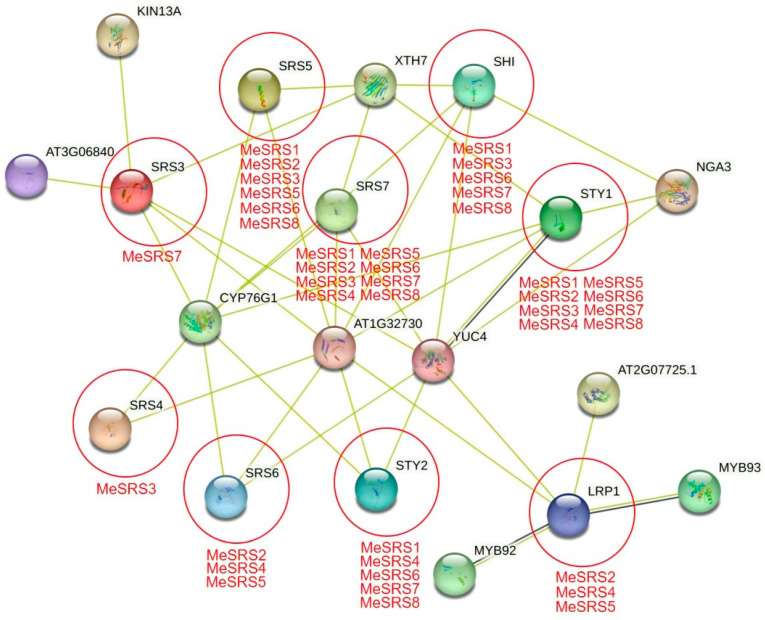
Protein–protein interaction network of MeSRS proteins based on their orthologs in *Arabidopsis*. The color scales represent the relative signal intensity scores.

**Figure 7 genes-14-00870-f007:**
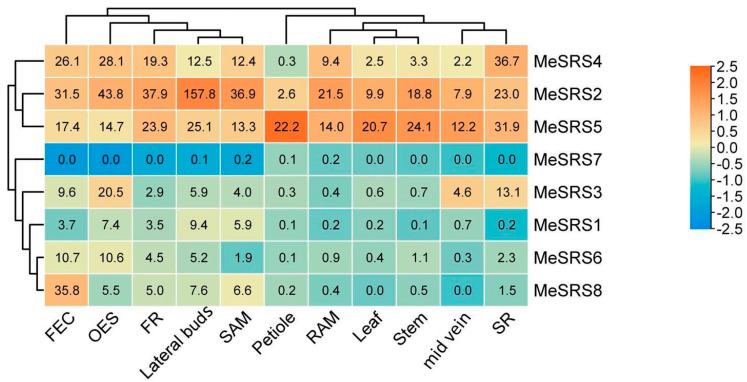
The expression profiles of *MeSRS* genes in various cassava tissues. Colors from red to blue indicate high to low expressions. FEC, OES, FR, SAM, RAM, and SR represent friable embryogenic callus, organized embryogenic structures, fibrous root, shoot apical meristem, root apical meristem, and storage root, respectively.

**Figure 8 genes-14-00870-f008:**
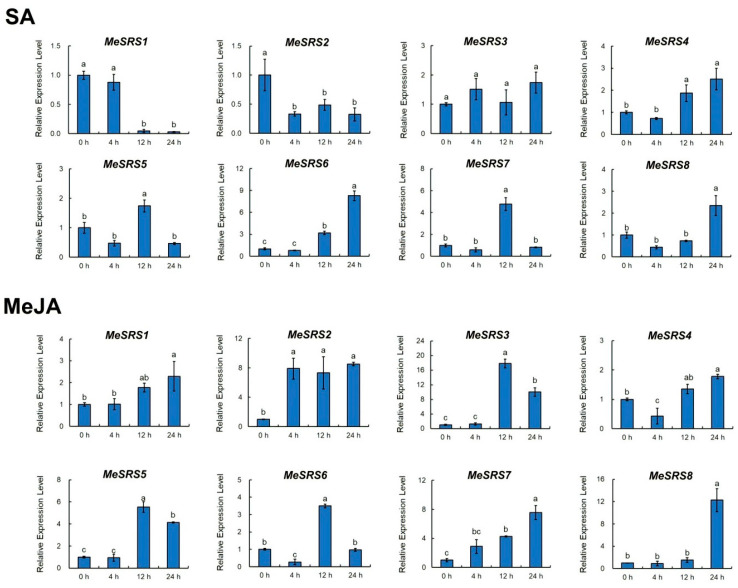
Expression profiles of *MeSRS* genes under phytohormone treatment in cassava as determined by qRT-PCR. The error bars represent the standard error of the means of three independent replicates. Values denoted by the same letter did not differ significantly at *p* < 0.05 according to Duncan’s multiple range tests.

**Figure 9 genes-14-00870-f009:**
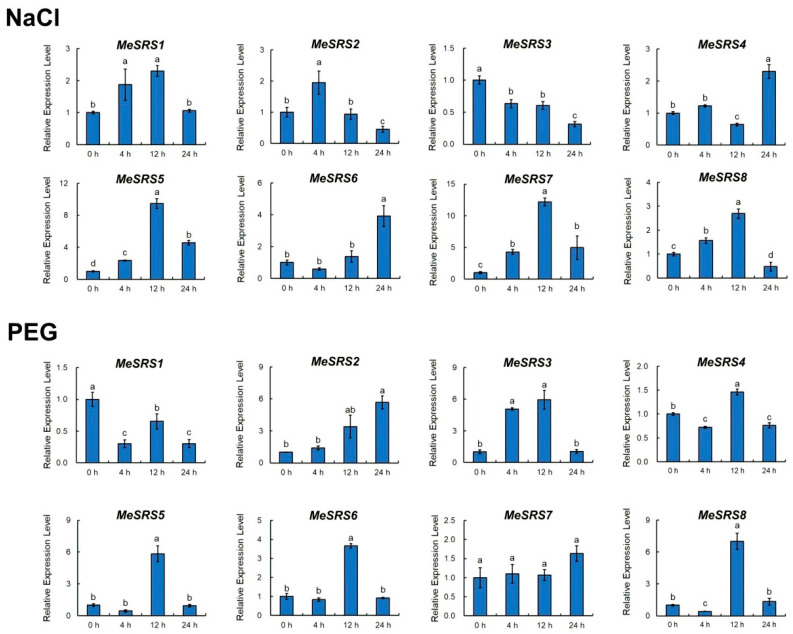
Expression profiles of *MeSRS* genes under abiotic stress treatment in cassava as determined by qRT-PCR. The error bars represent the standard error of the means of three independent replicates. Values denoted by the same letter did not differ significantly at *p* < 0.05 according to Duncan’s multiple range tests.

**Table 1 genes-14-00870-t001:** Basic information on SRS family in cassava.

Gene Name	Transcript Name	Location Coordinates (5′–3′)	Full CDS Length (bp)	Protein				Sublocation (WoLF)
				Protein Length (a.a.)	MW (kDa)	*p*I	GRAVY	
*MeSRS1*	Manes.01G141500.1	Chr01:25446260..25447722 forward	1023	340	36.34	7.53	−0.60	Nucleus
*MeSRS2*	Manes.01G179500.1	Chr01:27965586..27966905 forward	1080	359	37.22	8.71	−0.56	Endoplasmic reticulum
*MeSRS3*	Manes.02G100100.1	Chr02:7467865..7470885 forward	999	332	35.51	6.44	−0.55	Nucleus
*MeSRS4*	Manes.02G140000.1	Chr02:10395124..10397959 forward	1056	351	36.50	7.23	−0.53	Nucleus
*MeSRS5*	Manes.04G159200.1	Chr04:28097243..28098912 reverse	804	267	29.00	7.56	−0.71	Nucleus
*MeSRS6*	Manes.05G153200.1	Chr05:21854523..21857069 forward	909	302	33.40	6.13	−0.79	Nucleus
*MeSRS7*	Manes.11G087500.1	Chr11:12235314..12236199 reverse	642	213	23.05	8.98	−0.35	Nucleus
*MeSRS8*	Manes.18G018200.1	Chr18:1442270..1444681 forward	972	323	35.97	8.93	−0.79	Nucleus

**Table 2 genes-14-00870-t002:** The duplication events of *MeSRS* genes identified in cassava.

Sequence 1	Sequence 2	Ka	Ks	Ka/Ks	Duplication Type
*MeSRS1*	*MeSRS8*	0.232	1.614	0.144	Segmental
*MeSRS1*	*MeSRS3*	0.071	0.299	0.238	Segmental
*MeSRS1*	*MeSRS6*	0.251	1.964	0.128	Segmental
*MeSRS2*	*MeSRS4*	0.060	0.353	0.169	Segmental
*MeSRS3*	*MeSRS6*	0.268	1.919	0.140	Segmental
*MeSRS7*	*MeSRS8*	0.358	1.246	0.287	Segmental
*MeSRS8*	*MeSRS3*	0.276	1.306	0.211	Segmental
*MeSRS8*	*MeSRS6*	0.074	0.272	0.273	Segmental

## Data Availability

Not applicable.

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
