# Peer review of "Genome-Wide Identification and Expression Analysis of the SHI-Related Sequence Family in Cassava"

_genes, 2023, doi:10.3390/genes14040870_

Round 1
Reviewer 1 Report
Dear Authors,
Suggestions were highlighted on the text and notes were added.
Best regards.

Author Response
Thank you for your comments concerning our manuscript. We have studied comments carefully and have made correction which we hope meet with approval. Revised portions are marked red in the new manuscript. The main corrections in the paper and the responds to your comments are as flowing:
Line 39, 43, 63, 439: We are sorry for our carelessness and all the errors have fixed. The information of reference shown in revised manuscript.
Line 82: The “influence” has been revised to “influencing”.
Line 85: The Latin name “Lotus japonicas” has been revised to “Lotus japonicas (Regel) K.Larsen”.
Line 117: The “physiochemical” has been revised to “physicochemical”.
Line 129-130: The “random coil” has been revised to “the random coil”.
Line 137: The “sequences” has been revised to “sequence”.
Line 148: The “member” has been revised to “members”.
Line 154: The “Manihot esculanta ” has been revised to “Manihot esculanta Crantz”.
Line 181: The “value” has been revised to “values”.
Line 190: The “colinear” has been revised to “collinear”.
Line 199: The “orthologue” has been revised to “orthologous”.
Line 205: The “O. sativa detec” has been revised to “O. Sativa to detec”.
Line 229: The “stressor responses” has been revised to “stress responses”.
Line 231: The “element” has been revised to “elements”.
Line 239: The “using” has been revised to “using the”.
Line 261: The “component” has been revised to “components”.
Line 274: The “bud” has been revised to “buds”.
Line 358: The “conversed” has been revised to “conserved”.
Line 379: The “phylogenic tree” has been revised to “phylogentic tree”.
Line 431: The “Phytomoze” has been revised to “Phytozome”.
Line452: The “web server” has been revised to “web servers”.
Line481: The “midveins” has been revised to “mid veins”.
Line 514: The “to” has been revised to “to the”.
Line: The part of the Material and Methods should be positioned behind discussion according to the submission guidelines of the journal.
Furthermore, all words related have been checked and revised carefully, and the manuscript also has been standard editing revised.

Reviewer 2 Report
Overall, this research article represents an interesting investigation on “Genome-wide identification and expression analysis of the SHI-related sequence family in cassava”. Abstract seems logical and providing the concise summary of the findings, The introduction provides sufficient background of the study, the methods are generally appropriate for the experiments conducted. The analysis and results presented in figures seem logical while interpretation is supported by results. Moreover, the results are clearly described making the manuscript easily understandable for readers. In order to improve the present study, some essential modifications have to be fixed before it proceeds, and decisive action can be taken. In addition, the study needs basic editing on language and grammatical issues. All the comments and remarks are given below.
Discuss the limitations of the previous works as a motivation of the current study.
In abstract, when writing shortform for the first time, please write its full form as well, e.g., “SHI” in first line of abstract.
In material and methods, add some more description about the cultivar used in present study.
Line 470, please double check the sentence.
Some of the references are very old, please update the references.
The quality of figure 3D needs to be improved.
Please update the references as some of in-text citations are showing this error “[Error! Bookmark not defined.]”.
Please add reference to this “The SHORT INTERNODES (SHI) - related sequence (SRS) gene family, also termed as the SHI gene family or the Stylish (STY) family, is a plant-specific transcription factor family that encodes proteins with single zinc finger motifs.”
Please update this information, as 2020 has passed long ago. “The Food and Agriculture Organization (FAO) estimated that by 2020, 302.6 million tons of cassava would be produced globally on approximately 28.24 million hectares”
There are numerous mistakes in English language and grammar.
Author Response
Thank you for your comments concerning our manuscript. We have studied comments carefully and have made correction which we hope meet with approval. Revised portions are marked red in the new manuscript. The main corrections in the paper and the responds to your comments are as flowing:
Point 1: Discuss the limitations of the previous works as a motivation of the current study.
Response 1: We have added the sentences “SRS is a plant TF family which has an important regulatory function in plant development and growth. In A. thaliana, SRS family genes have well been demonstrated to play vital roles in regulating the photomorphogenesis [21,22], floral organ development [14,18], lateral root formation [17,21], carpel development [15], and phytohormone response [10,16]. Recent researches have further confirmed the involvement of SRS genes in abiotic stresses and phytohormone response [17,27,28,32], and has potential utilization value for genetic improvement of crop resistance. Although researches had been performed to SRS genes in several plant species [17,27-31], no systematic research on the SRS gene family in cassava has been reported. It is necessary to investigate the possible role of SRS in cassava” in Line 324-333.
Point 2: In abstract, when writing shortform for the first time, please write its full form as well, e.g., “SHI” in first line of abstract.
Response 2: We have revised such words in abstract, please see Line 11.
Point 3: In material and methods, add some more description about the cultivar used in present study.
Response 3:We've added it after line488-489.
South China 9 (SC9) cassava, a common edible cultivar in China that has good performance at properties of starch and stress tolerance, was used in this research.
Point 4: Line 470, please double check the sentence.
Response 4: Revised. RNAprep Pure Plan Plus Kit (TIANGEN Biotech Co., Ltd., Beijing) was used to isolate RNA from the leaves according to the manufacturer’s instructions. Total RNA (1 ug) was used for first-strand cDNA synthesis by a reverse transcriptase kit (M1631, Thermo, USA).
Point 5: Some of the references are very old, please update the references.
Response 5: We have replaced several old references. The new references are marked red.
Point 6: The quality of figure 3D needs to be improved.
Response 6: The figure 3D has been improved.
Point 7: Please update the references as some of in-text citations are showing this error “[Error! Bookmark not defined.]”.
Response 7: We are sorry for our carelessness and all the errors have fixed.
Point 8: Please add reference to this “The SHORT INTERNODES (SHI) - related sequence (SRS) gene family, also termed as the SHI gene family or the Stylish (STY) family, is a plant-specific transcription factor family that encodes proteins with single zinc finger motifs.”
Response 8: Thanks for your suggestion. The reference published by Fridborg et al. (1999) has been added in Line 59.
Reference:
Fridborg, I., Kuusk, S., Moritz, T., and Sundberg, E. (1999). The Arabidopsis dwarf mutant shi exhibits reduced gibberellin responses conferred by overexpression of a new putative zinc finger protein. Plant Cell 11, 1019–1031.
Point 9: Please update this information, as 2020 has passed long ago. “The Food and Agriculture Organization (FAO) estimated that by 2020, 302.6 million tons of cassava would be produced globally on approximately 28.24 million hectares”
Response 9: Thanks for your suggestion. The latest figure about global cassava production has updated to 2021. We have updated this information and rewritten the sentence as “The Food and Agriculture Organization (FAO) estimated that by 2021, 306 million tons of cassava would be produced globally on approximately 34.23 million hectares” in line40-42
Furthermore, all words related have been checked and revised carefully, and the manuscript also has been standard editing revised.
